# Learning to Learn Graph Topologies

**Xingyue Pu**
University of Oxford
xpu@robots.ox.ac.uk

**Tianyue Cao    Xiaoyun Zhang**
Shanghai Jiao Tong University
{vanessa_,xiaoyun.zhang}@sjtu.edu.cn

**Xiaowen Dong**
University of Oxford
xdong@robots.ox.ac.uk

**Siheng Chen**[*]
Shanghai Jiao Tong University
sihengc@sjtu.edu.cn

## Abstract

Learning a graph topology to reveal the underlying relationship between data entities plays an important role in various machine learning and data analysis tasks. Under the assumption that structured data vary smoothly over a graph, the problem can be formulated as a regularised convex optimisation over a positive semidefinite cone and solved by iterative algorithms. Classic methods require an explicit convex function to reflect generic topological priors, e.g. the $\ell_1$ penalty for enforcing sparsity, which limits the flexibility and expressiveness in learning rich topological structures. We propose to learn a mapping from node data to the graph structure based on the idea of learning to optimise (L2O). Specifically, our model first unrolls an iterative primal-dual splitting algorithm into a neural network. The key structural proximal projection is replaced with a variational autoencoder that refines the estimated graph with enhanced topological properties. The model is trained in an end-to-end fashion with pairs of node data and graph samples. Experiments on both synthetic and real-world data demonstrate that our model is more efficient than classic iterative algorithms in learning a graph with specific topological properties.

## 1  Introduction

Graphs are an effective modelling language for revealing relational structure in high-dimensional complex domains and may assist in a variety of machine learning tasks. However, in many practical scenarios an explicit graph structure may not be readily available or easy to define. Graph learning aims at learning a graph topology from observation on data entities and is therefore an important problem studied in the literature.

Model-based graph learning from the perspectives of probabilistic graphical models [24] or graph signal processing [32, 30, 28] solves an optimisation problem over the space of graph candidates whose objective function reflects the inductive graph-data interactions. The data are referred as to the observations on nodes, node features or graph signals in literature. The convex objective function often contains a graph-data fidelity term and a structural regulariser. For example, by assuming that data follow a multivariate Gaussian distribution, the graphical Lasso [2] maximises the $\ell_1$ regularised log-likelihood of the precision matrix which corresponds to a conditional independence graph. Based on the convex formulation, the problem of learning an optimal graph can be solved by many iterative algorithms, such as proximal gradient descent [15], with convergence guarantees.

These classical graph learning approaches, despite being effective, share several common limitations. Firstly, handcrafted convex regularisers may not be expressive enough for representing the

---

[*]Corresponding author

35th Conference on Neural Information Processing Systems (NeurIPS 2021).

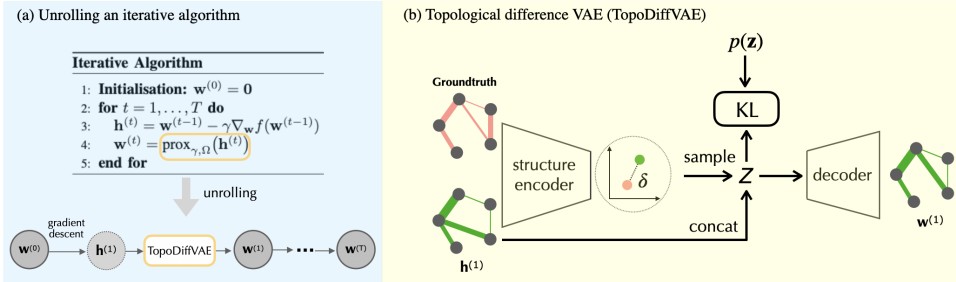

Figure 1: An illustration of the proposed framework.

rich topological and structural priors. In the literature, only a limited number of graph structural properties can be modelled explicitly using convex regularisers, e.g. the $\ell_1$ penalty for enforcing sparsity. More complex structures such as scale-free and small-world properties have been largely overlooked due to the difficulty in imposing them via simple convex optimisation. Secondly, graph structural regularisers might not be differentiable despite being convex, which perplexed the design of optimisation algorithms. Thirdly, iterative algorithms might take long to converge, as the search space grows quadratically with the graph size. Fourthly, tuning penalty parameters in front of structural regularisers and the step size of iterative algorithms are laborious.

To address the above limitations, we propose a novel functional learning framework to learn a mapping from node observations to the underlying graph topology with desired structural property. Our framework is inspired by the emerging field of *learning to optimise* (L2O) [8, 26]. Specifically, as shown in Figure 1, we first unroll an iterative algorithm for solving the aforementioned regularised graph learning objective. To further increase the flexibility and expressiveness, we design a topological difference variational autoencoder (TopoDiffVAE) to replace the proximal projector of structural regularisation functions in the original iterative steps. We train the model in an end-to-end fashion with pairs of data and graphs that share the same structural properties. Once trained, the model can be deployed to learn a graph topology that exhibits such structural properties.

The proposed method learns topological priors from graph samples, which are more expressive in learning structured graphs compared to traditional methods using analytic regularisers. Compared to deep neural networks, unrolling an iterative algorithm that is originally designed for graph learning problem introduces a sensible inductive bias that makes our model highly interpretable. We test the effectiveness and robustness of the proposed method in learning graphs with diverse topological structures on both synthetic and real-world data.

The main contributions are as follows. Firstly, we propose a novel and highly interpretable neural networks to learn a data-to-graph mapping based on algorithmic unrolling. Secondly, we propose an effective TopoDiffVAE module that replaces the proximal operators of structural regularisers. To the best of our knowledge, this constitutes the first attempt in addressing the difficult challenge of designing convex functions to represent complex topological priors, which in turn allows for improved flexibility and expressiveness. Finally, the proposed method improves the accuracy of graph learning by 80% compared to traditional iterative solvers.

## 2    Problem Formulation

**Model-based graph learning**    Let $G = \{\mathcal{V}, \mathcal{E}, \mathbf{W}\}$ be an undirected weighted graph, where $\mathcal{V}$ is the node set with $|\mathcal{V}| = m$, $\mathcal{E}$ is the edge set and $\mathbf{W}$ is the weighted adjacency matrix whose $ij$-th entry embodies the similarity between node $i$ and node $j$. The combinatorial graph Laplacian matrix is defined as $\mathbf{L} = \mathbf{D} - \mathbf{W}$, where $\mathbf{D} = \mathrm{diag}(\mathbf{W1})$ is a diagonal matrix of node degrees. In many scenarios, we have access to a structured data matrix, which is denoted as $\mathbf{X} = [\mathbf{x}_1, \mathbf{x}_2, \ldots, \mathbf{x}_m]^\top \in \mathbb{R}^{m \times n}$, where each column $\mathbf{x}_i$ can be considered as a signal on the graph $G$. The goal of graph learning is to infer such a graph $G$ that is fully represented by $\mathbf{L}$ or $\mathbf{W}$ from $\mathbf{X}$. Specifically, we solve a generic optimisation problem

$$\min_{\mathbf{L}} \mathrm{tr}(\mathbf{X}^\top \mathbf{L} \mathbf{X}) + \Omega(\mathbf{L}), \tag{1}$$

where the first term is the so-called Laplacian quadratic form, measuring the variation of $\mathbf{X}$ on the graph, and $\Omega(\mathbf{L})$ is a convex regulariser on $\mathbf{L}$ to reflect structural priors. Since $\mathbf{W}$ is symmetric, we introduce the vector of edge weights $\mathbf{w} \in \mathbb{R}_+^{m(m-1)/2}$ and reparameterise Eq.(1) based on $\text{tr}(\mathbf{X}^\top \mathbf{L} \mathbf{X}) = \sum_{i,j} \mathbf{W}_{ij} ||\mathbf{x}_i - \mathbf{x}_j||_2^2 = 2\mathbf{w}^\top \mathbf{y}$, where $\mathbf{y} \in \mathbb{R}_+^{m(m-1)/2}$ is the half-vectorisation of the Euclidean distance matrix. Now, we optimise the objective w.r.t $\mathbf{w}$ such that

$$\min_{\mathbf{w}} 2\mathbf{w}^\top \mathbf{y} + \mathcal{I}_{\{\mathbf{w} \geq 0\}}(\mathbf{w}) + \Omega_1(\mathbf{w}) + \Omega_2(\mathcal{D}\mathbf{w}), \tag{2}$$

where $\mathcal{I}$ is an indicator function such that $\mathcal{I}_\mathcal{C}(\mathbf{w}) = 0$ if $\mathbf{w} \in \mathcal{C}$ and $\mathcal{I}_\mathcal{C}(\mathbf{w}) = \infty$ otherwise, and $\mathcal{D}$ is a linear operator that transforms $\mathbf{w}$ into the vector of node degrees such that $\mathcal{D}\mathbf{w} = \mathbf{W1}$. The reparameterisation reduces the dimension of search space by a half and we do not need to take extra care of the positive semi-definiteness of $\mathbf{L}$.

In Eq.(2), the regularisers $\Omega$ is split into $\Omega_1$ and $\Omega_2$ on edge weights and node degrees respectively. Both formulations can be connected to many previous works, where regularisers are mathematically handcrafted to reflect specific graph topologies. For example, the $\ell_1$-regularised log-likelihood maximisation of the precision matrix with Laplacian constraints [23, 22] can be seen as a special case of Eq.(1), where $\Omega(\mathbf{L}) = \log \det(\mathbf{L} + \sigma^2 \mathbf{I}) + \lambda \sum_{i \neq j} |\mathbf{L}_{ij}|$. The $\ell_1$ norm on edge weights enforces the learned conditional independence graph to be sparse. The author in [16] considers the log-barrier on node degrees, i.e. $\Omega_2(\mathcal{D}\mathbf{w}) = -\mathbf{1}^\top \log(\mathcal{D}\mathbf{w})$, to prevent the learned graph with isolated nodes. Furthermore, $\Omega_1(\mathbf{w}) = ||\mathbf{w}||_2^2$ is often added to force the edge weights to be smooth [11, 16].

**Learning to learn graph topologies** In many real-world applications, the mathematically-designed topological priors in the above works might not be expressive enough. For one thing, many interesting priors are too complex to be abstracted as an explicit convex function, e.g. the small-world property [36]. Moreover, those constraints added to prevent trivial solutions might conflict with the underlying structure. For example, the aforementioned log barrier may encourage many nodes with a large degree, which contradicts the power-law assumption in scale-free graphs. The penalty parameters, treated as a part of regularisation, are manually appointed or tuned from inadequately granularity, which may exacerbate the lack of topological expressiveness. These motivate us to directly learn the topological priors from graphs.

Formally, we aim at learning a neural network $\mathcal{F}_{\boldsymbol{\theta}}(\cdot)$ that maps the data term $\mathbf{y}$ to a graph representation $\mathbf{w}$ that share the same topological properties and hence belong to the same graph family $\mathcal{G}$. With training pairs $\{(\mathbf{y}_i, \mathbf{w}_i)\}_{i=1}^n$, we consider a supervised learning framework to obtain optimal $\mathcal{F}_{\boldsymbol{\theta}}^*$ such that

$$\boldsymbol{\theta}^* = \arg\min_{\boldsymbol{\theta}} \mathbb{E}_{\mathbf{w} \sim \mathcal{G}}[\mathcal{L}(\mathcal{F}_{\boldsymbol{\theta}}(\mathbf{y}), \mathbf{w})], \tag{3}$$

where $\mathcal{L}(\widehat{\mathbf{w}}, \mathbf{w})$ is a loss surrogate between the estimation $\widehat{\mathbf{w}} = \mathcal{F}_{\boldsymbol{\theta}}(\mathbf{y})$ and the groundtruth. Once trained, the optimal $\mathcal{F}_{\boldsymbol{\theta}}^*(\cdot)$ implicitly carries topological priors and can be applied to estimate graphs from new observations. Therefore, our framework promotes learning to learn a graph topology. In the next section, we elaborate on the architecture of $\mathcal{F}_{\boldsymbol{\theta}}(\cdot)$ and the design of $\mathcal{L}(\cdot, \cdot)$.

## 3 Unrolled Networks with Topological Enhancement

In summary, the data-to-graph mapping $\mathcal{F}_{\boldsymbol{\theta}}(\cdot)$ is modelled with unrolling layers (Section 3.1) inherited from a primal-dual splitting algorithm that is originally proposed to solve the model-based graph learning. We replace the proximal operator with a trainable variational autoencoder (i.e. TopoDiffVAE) that refines the graph estimation with structural enhancement in Section 3.2. The overall network is trained in an end-to-end fashion with the loss function introduced in Section 3.3.

### 3.1 Unrolling primal-dual iterative algorithm

Our model leverages the inductive bias from model-based graph learning. Specifically, we consider a special case of the formulation in Eq. (2) such that

$$\min_{\mathbf{w}} 2\mathbf{w}^\top \mathbf{y} + \mathcal{I}_{\{\mathbf{w} \geq 0\}}(\mathbf{w}) - \alpha \mathbf{1}^\top \log(\mathcal{D}\mathbf{w}) + \beta ||\mathbf{w}||_2^2 \tag{4}$$

where $\Omega_1(\mathbf{w}) = \beta||\mathbf{w}||_2^2$ and $\Omega_2(\mathcal{D}\mathbf{w}) = -\alpha\mathbf{1}^\top\log(\mathcal{D}\mathbf{w})$, both of which are mathematically handcrafted topological priors. To solve this optimisation problem, we consider a forward-backward-forward primal-dual splitting algorithm (PDS) in Algorithm 1. The algorithm is specifically derived from Algorithm 6 in [21] by the authors in [16]. In Algorithm 1, Step 3, 5 and 7 update the primal variable that corresponds to $\mathbf{w}$, while Step 4, 6 and 8 update the dual variable in relation to node degrees. The forward steps are equivalent to a gradient descent on the primal variable (Step 3 and 7) and a gradient ascent on the the dual variable (Step 4 and 8), both of which are derived from the differentiable terms in Eq.(4). The backward steps (Step 5 and 6) are proximal projections that correspond to two non-differentiable topological regularisers, i.e. the non-negative constraints on $\mathbf{w}$ and the log barrier on node degrees in Eq.(4). The detailed derivations are attached in Appendix A.2. It should be note that $\ell_1$ norm that promotes sparsity is implicitly included, i.e. $2\mathbf{w}^\top\mathbf{y} = 2\mathbf{w}^\top(\mathbf{y} - \lambda/2) + \lambda||\mathbf{w}||_1$ given $\mathbf{w} \geq 0$. We choose to unroll PDS as the main network architecture of $\mathcal{F}_{\boldsymbol{\theta}}(\cdot)$, since the separate update steps of primal and dual variables allow us to replace of proximal operators that are derived from other priors.

---

**Algorithm 1** PDS

**Input:** $\mathbf{y}, \gamma, \alpha$ and $\beta$.
 1: **Initialisation:** $\mathbf{w}^{(0)} = \mathbf{0}, \mathbf{v}^{(0)} = \mathbf{0}$
 2: **while** $|\mathbf{w}^{(t)} - \mathbf{w}^{(t-1)}| > \epsilon$ **do**
 3: $\quad \mathbf{r}_1^{(t)} = \mathbf{w}^{(t)} - \gamma(2\beta\mathbf{w}^{(t)} + 2\mathbf{y} + \mathcal{D}^\top\mathbf{v}^{(t)})$
 4: $\quad \mathbf{r}_2^{(t)} = \mathbf{v}^{(t)} + \gamma\mathcal{D}\mathbf{w}^{(t)}$
 5: $\quad \mathbf{p}_1^{(t)} = \text{prox}_{\gamma,\Omega_1}(\mathbf{r}_1^{(t)}) = \max\{\mathbf{0}, \mathbf{r}_1^{(t)}\}$

 6: $\quad \mathbf{p}_2^{(t)} = \text{prox}_{\gamma,\Omega_2}(\mathbf{r}_2^{(t)})$, where

$\quad \left(\text{prox}_{\gamma^{(t)},\Omega_2}(\mathbf{r}_2)\right)_i = \frac{r_{2,i} - \sqrt{r_{2,i}^2 + 4\alpha\gamma}}{2}$
 7: $\quad \mathbf{q}_1^{(t)} = \mathbf{p}_1^{(t)} - \gamma(2\beta\mathbf{p}_1^{(t)} + 2\mathbf{y} + \mathcal{D}^\top\mathbf{p}_2^{(t)})$
 8: $\quad \mathbf{q}_2^{(t)} = \mathbf{p}_2^{(t)} + \gamma\mathcal{D}\mathbf{p}_1^{(t)}$
 9: $\quad \mathbf{w}^{(t+1)} = \mathbf{w}^{(t)} - \mathbf{r}_1^{(t)} + \mathbf{q}_1^{(t)}$
10: $\quad \mathbf{v}^{(t+1)} = \mathbf{v}^{(t)} - \mathbf{r}_2^{(t)} + \mathbf{q}_2^{(t)}$
11: **end while**
12: **return** $\widehat{\mathbf{w}} = \mathbf{w}^{(T)}$

---

**Algorithm 2** Unrolling Net (L2G)

**Input:** $\mathbf{y}, T, \texttt{enhancement}^{(t)} \in \{\texttt{TRUE, FALSE}\}$
 1: **Initialisation:** $\mathbf{w}^{(0)} = \mathbf{0}, \mathbf{v}^{(0)} = \mathbf{0}$
 2: **for** $t = 0, 1, \ldots, T$ **do**
 3: $\quad \mathbf{r}_1^{(t)} = \mathbf{w}^{(t)} - \gamma^{(t)}(2\beta^{(t)}\mathbf{w}^{(t)} + 2\mathbf{y} + \mathcal{D}^\top\mathbf{v}^{(t)})$
 4: $\quad \mathbf{r}_2^{(t)} = \mathbf{v}^{(t)} + \gamma^{(t)}\mathcal{D}\mathbf{w}^{(t)}$
 5: $\quad$ if $\texttt{enhancement}^{(t)}$=TRUE,
 $\quad\quad \mathbf{p}_1^{(t)} = \text{TopoDiffVAE}(\mathbf{r}_1^{(t)});$
 $\quad$ else,
 $\quad\quad \mathbf{p}_1^{(t)} = \text{prox}_{\gamma,\Omega_1}(\mathbf{r}_1^{(t)}) = \max\{\mathbf{0}, \mathbf{r}_1^{(t)}\}.$
 6: $\quad \mathbf{p}_2^{(t)} = \text{prox}_{\gamma^{(t)},\Omega_2}(\mathbf{r}_2^{(t)})$, where

$\quad \left(\text{prox}_{\gamma^{(t)},\Omega_2}(\mathbf{r}_2)\right)_i = \frac{r_{2,i} - \sqrt{r_{2,i}^2 + 4\alpha^{(t)}\gamma^{(t)}}}{2}$
 7: $\quad \mathbf{q}_1^{(t)} = \mathbf{p}_1^{(t)} - \gamma^{(t)}(2\beta^{(t)}\mathbf{p}_1^{(t)} + 2\mathbf{y} + \mathcal{D}^\top\mathbf{p}_2^{(t)})$
 8: $\quad \mathbf{q}_2^{(t)} = \mathbf{p}_2^{(t)} + \gamma^{(t)}\mathcal{D}\mathbf{p}_1^{(t)}$
 9: $\quad \mathbf{w}^{(t+1)} = \mathbf{w}^{(t)} - \mathbf{r}_1^{(t)} + \mathbf{q}_1^{(t)}$
10: $\quad \mathbf{v}^{(t+1)} = \mathbf{v}^{(t)} - \mathbf{r}_2^{(t)} + \mathbf{q}_2^{(t)}$
11: **end for**
12: **return** $\mathbf{w}^{(1)}, \mathbf{w}^{(2)}, \ldots, \mathbf{w}^{(T)} = \widehat{\mathbf{w}}$

---

The proposed unrolling network is summarised in Algorithm 2, which unrolls the updating rules of Algorithm 1 for $T$ times and is hypothetically equivalent to running the iteration steps for $T$ times. In Algorithm 2, we consider two major substitutions to enable more flexible and expressive learning.

Firstly, we replace the hyperparameters $\alpha$, $\beta$ and the step size $\gamma$ in the PDS by layer-wise trainable parameters (highlighted in blue in Algorithm 2), which can be updated through backpropagation. Note that we consider independent trainable parameters in each unrolling layer, as we found it can enhance the representation capacity and boost the performance compared to the shared unrolling scheme [14, 26].

Secondly, we provide a layer-wise optional replacement for the primal proximal operator in Step 5. When learning complex topological priors, a trainable neural network, i.e. the TopoDiffVAE (highlighted in red in Algorithm 2) takes the place to enhance the learning ability. TopoDiffVAE will be defined in Section 3.2. It should be noted that such a replacement at every unrolling layer might lead to overly complex model that is hard to train and might suffer from overfitting, which depends on the graph size and number of training samples. Empirical results suggest TopoDiffVAE takes effects mostly on the last few layers. Comprehensibly, at first several layers, L2G quickly search a near-optimal solution with the complete inductive bias from PDS, which provides a good initialisation for TopoDiffVAE that further promotes the topological properties. In addition, we do not consider a trainable function to replace the proximal operator in Step 6 of Algorithm 1, as it

projects dual variables that are associated to node degrees but do not draw a direct equality. It is thus difficult to find a loss surrogate and a network architecture to approximate such an association.

### 3.2 Topological difference variational autoencoder (TopoDiffVAE)

The proximal operator in Step 5 of Algorithm 1 follows from $\Omega_1(\mathbf{w})$ in Eq. (4). Intuitively, an ideal regulariser in Eq. (4) would be an indicator function $\Omega_1 = \mathcal{I}_\mathcal{G}(\cdot)$, where $\mathcal{G}$ is a space of graph sharing same topological properties. The consequent proximal operator projects $\mathbf{w}$ onto $\mathcal{G}$. However, it is difficult to find such an oracle function. To adaptively learn topological priors from graph samples and thus achieve a better expressiveness in graph learning, we design a topological difference variational autoencoder (TopoDiffVAE) to replace the handcrafted proximal operators in Step 5 of Algorithm 1.

Intuitively, the goal is to learn a mapping between two graph domains $\mathcal{X}$ and $\mathcal{Y}$. The graph estimate after gradient descent step $\mathbf{r}_1^{(t)} \in \mathcal{X}$ lacks topological properties while the graph estimate after this mapping $\mathbf{p}_1^{(t)} \in \mathcal{Y}$ is enhanced with the topological properties that are embodied in the groundtruth $\mathbf{w}$, e.g. graphs without and with the scale-free property. We thus consider a conditional VAE such that $\mathcal{P} : (\mathbf{r}_1^{(t)}, \mathbf{z}) \to \mathbf{w}$, where the latent variable $\mathbf{z}$ has an implication of the topological difference between $\mathbf{r}_1^{(t)}$ and $\mathbf{w}$. In Algorithm 2, $\mathbf{p}_1^{(t)}$ is the estimate of the groundtruth $\mathbf{w}$ from $\mathcal{P}$. When we talk about the topological difference, we mainly refer to binary structure. This is because edge weights might conflict with the topological structure, e.g. an extremely large edge weight of a node with only one neighbour in a scale-free network. We want to find a continuous representation $\mathbf{z}$ for such discrete and combinatorial information that allows the backpropagation flow. A straightforward solution is to embed both graph structures with graph convolutions networks and compute the differences in the embedding space. Specifically, the encoder and decoder of the TopoDiffVAE $\mathcal{P}$ are designed as follows.

**Encoder.** We extract the binary adjacency matrix of $\mathbf{r}_1^{(t)}$ and $\mathbf{w}$ by setting the edge weights less than $\eta$ to zero and those above $\eta$ to one. Empirically, $\eta$ is a small value to exclude the noisy edge weights, e.g. $\eta = 10^{-4}$. The binary adjacency matrix are denoted as $\mathbf{A}_r$ and $\mathbf{A}_w$ respectively. Following [19], the approximate posterior $q_\phi(\mathbf{z}|\mathbf{r}_1^{(t)}, \mathbf{w})$ is modelled as a Gaussian distribution whose mean $\boldsymbol{\mu}$ and covariance $\boldsymbol{\Sigma} = \boldsymbol{\sigma}^2\mathbf{I}$ are transformed from the topological difference between the embedding of $\mathbf{A}_r$ and $\mathbf{A}_w$ such that

$$\boldsymbol{\delta} = f_{\text{emb}}(\mathbf{A}_w) - f_{\text{emb}}(\mathbf{A}_r) \tag{5a}$$

$$\boldsymbol{\mu} = f_{\text{mean}}(\boldsymbol{\delta}), \quad \boldsymbol{\sigma} = f_{\text{cov}}(\boldsymbol{\delta}), \quad \mathbf{z}|\mathbf{r}_1^{(t)}, \mathbf{w} \sim \mathcal{N}(\boldsymbol{\mu}, \boldsymbol{\sigma}^2\mathbf{I}) \tag{5b}$$

where $f_{\text{mean}}$ and $f_{\text{cov}}$ are two multilayer perceptrons (MLPs). The embedding function $f_{\text{emb}}$ is a 2-layer graph convolutional network[20] followed by a readout layer $f_{\text{readout}}$ that averages the node representation to obtain a graph representation, that is,

$$f_{\text{emb}}(\mathbf{A}) = f_{\text{readout}}\Big(\psi\big(\mathbf{A}\psi(\mathbf{A}\mathbf{d}\mathbf{h}_0^\top)\mathbf{H}_1\big)\Big), \tag{6}$$

where $\mathbf{A}$ refers to either $\mathbf{A}_r$ and $\mathbf{A}_w$, $\mathbf{h}_0$ and $\mathbf{H}_1$ are learnable parameters at the first and second convolution layer that are set as the same in embedding $\mathbf{A}_r$ and $\mathbf{A}_w$. $\psi$ is the non-linear activation function. Node degrees $\mathbf{d} = \mathbf{A}\mathbf{1}$ are taken as an input feature. This architecture is flexible to incorporate additional node features by simply replacing $\mathbf{d}$ with a feature matrix $\mathbf{F}$.

**Latent variable sampling.** We use Gaussian reparameterisation trick [19] to sample $\mathbf{z}$ such that $\mathbf{z} = \boldsymbol{\mu} + \boldsymbol{\sigma} \odot \boldsymbol{\epsilon}$, where $\boldsymbol{\epsilon} \sim \mathcal{N}(\mathbf{0}, \mathbf{I})$. We also restrict $\mathbf{z}$ to be a low-dimensional vector to prevent the model from ignoring the original input in the decoding step and degenerating into an auto-encoder. Meanwhile, the prior distribution of the latent variable is regularised to be close to a Gaussian prior $\mathbf{z} \sim \mathcal{N}(\mathbf{0}, \mathbf{I})$ to avoid overfitting. We minimise the Kullback–Leibler divergence between $q_\phi(\mathbf{z}|\mathbf{r}_1^{(t)}, \mathbf{w})$ and the prior $p(\mathbf{z})$ during training as shown in Eq.(8).

**Decoder.** The decoder is trained to reconstruct the groundtruth graph representations $\mathbf{w}$ (the estimate is $\mathbf{p}_1^{(t)}$) given the input $\mathbf{r}_1^{(t)}$ and the latent code $\mathbf{z}$ that represents topological differences. Formally, we obtain an augmented graph estimation by concatenating $\mathbf{r}_1^{(t)}$ and $\mathbf{z}$ and then feeding it into a 2-layer

MLP $f_{\text{dec}}$ such that

$$\mathbf{p}_1^{(t)} = f_{\text{dec}}(\text{CONCAT}[\mathbf{r}_1^{(t)}, \mathbf{z}]). \tag{7}$$

### 3.3 An end-to-end training scheme

By stacking the unrolling module and TopoDiffVAE, we propose an end-to-end training scheme to learn an effective data-to-graph mapping $\mathcal{F}_{\boldsymbol{\theta}}$ that minimises the loss from both modules such that

$$\min_{\boldsymbol{\theta}} \; \mathbb{E}_{\mathbf{w} \sim \mathcal{G}} \left[ \sum_{t=1}^{T} \left\{ \tau^{T-t} \frac{||\mathbf{w}^{(t)} - \mathbf{w}||_2^2}{||\mathbf{w}||_2^2} + \beta_{\text{KL}} \text{KL}\Big(q_{\phi}(\mathbf{z}|\mathbf{r}_1^{(t)}, \mathbf{w})||\mathcal{N}(\mathbf{0}, \mathbf{I})\Big) \right\} \right] \tag{8}$$

where $\mathbf{w}^{(t)}$ is the output of the $t$-th unrolling layer and $\mathbf{r}_1^{(t)}$ is the output of Step 3 in Algorithm 2, $\tau \in (0, 1]$ is a loss-discounting factor added to reduce the contribution of the normalised mean squared error of the estimates from the first few layers. This is similar to the discounted cumulative loss introduced in [31]. The first term represents the graph reconstruction loss, while the second term is the KL regularisation term in the TopoDiffVAE module. When the trade-off hyper-parameter $\beta_{\text{KL}} > 1$, the VAE framework reduces to $\beta$-VAE [7] that encourages disentanglement of latent factors $\mathbf{z}$. The trainable parameters in the overall unrolling network $\mathcal{F}_{\boldsymbol{\theta}}$ are $\boldsymbol{\theta} = \{\boldsymbol{\theta}_{\text{unroll}}, \boldsymbol{\theta}_{\text{TopoDiffVAE}}\}$, where $\boldsymbol{\theta}_{\text{unroll}} = \{\alpha^{(0)}, \dots, \alpha^{(T-1)}, \beta^{(0)}, \dots, \beta^{(T-1)}, \gamma^{(0)}, \dots, \gamma^{(T-1)}\}$ and $\boldsymbol{\theta}_{\text{TopoDiffVAE}}$ includes the parameters in $f_{\text{emb}}$, $f_{\text{mean}}$, $f_{\text{cov}}$ and $f_{\text{dec}}$. The optimal mapping $\mathcal{F}_{\boldsymbol{\theta}}^{\star}$ is further used to learn graph topologies that are assumed to have the same structural properties as training graphs.

## 4 Related Works

**Graph Learning.** Broadly speaking there are two approaches to the problem of graph learning in the literature: model-based and learning-based approaches. The model-based methods solve an regularised convex optimisation whose objective function reflects the inductive graph-data interactions and regularisation that enforces embody structural priors [2, 9, 25, 13, 23, 33, 12, 10, 22, 16, 11, 5]. They often rely on iterative algorithms which require specific designs, including proximal gradient descent [15], alternating direction method of multiplier (ADMM) [6], block-coordinate decent methods [13] and primal-dual splitting [16]. Our model unrolls a primal-dual splitting as an inductive bias, and it allows for regularisation on both the edge weights and node degrees.

Recently, learning-based methods have been proposed to introduce more flexibility to model structural priors in graph learning. GLAD [31] unrolls an alternating minimisation algorithm for solving an $\ell_1$ regularised log-likelihood maximisation of the precision matrix. It also allows the hyperparameters of the $\ell_1$ penalty term to differ element-wisely, which further enhances the representation capacity. Our work differs from GLAD in that (1) we optimise over the space of weighted adjacency matrices that contain non-negative constraints than over the space of precision matrices; (2) we unroll a primal-dual splitting algorithm where the whole proximal projection step is replaced with TopoDiffVAE, while GLAD adds a step of learning element-wise hyperparameters. Furthermore, GLAD assumes that the precision matrices have a smallest eigenvalue of 1, which affects its accuracy on certain datasets. Deep-Graph [4] uses a CNN architecture to learn a mapping from empirical covariance matrices to the binary graph structure. We will show in later sections that a lack of task-specific inductive bias leads to an inferior performance in recovering structured graphs.

**Learning to optimise.** Learning to optimise (L2O) combines the flexible data-driven learning procedure and interpretable rule-based optimisation [8]. Algorithmic unrolling is one main approach of L2O [26]. Early development of unrolling was proposed to solve sparse and low-rank regularised problems and the main goal is to reduce the laborious iterations of hand engineering [14, 34, 27, 1]. Another approach is Play and Plug that plugs pre-trained neural networks in replacement of certain update steps in an iterative algorithm [29, 37]. Our framework is largely inspired by both approaches, and constitutes a first attempt in utilising L2O for the problem of graph learning.

## 5 Experiments

**General settings.** Random graphs are drawn from the Barabási-Albert (BA), Erdös-Rényi (ER), stochastic block model (SBM) and Watts–Strogatz (WS) model to have scale-free, random,

Table 1: GMSE* in graph reconstruction

| model/graph type | Scale-free (BA) | Random sparse (ER) | Community (SBM) | Small-world (WS) |
|---|---|---|---|---|
| **Iterative algorithm:** | | | | |
| ADMM | $0.4094 \pm .0120$ | $0.3547 \pm .0120$ | $0.3168 \pm .0226$ | $0.2214 \pm .0151$ |
| PDS | $0.4033 \pm .0072$ | $0.3799 \pm .0085$ | $0.3111 \pm .0147$ | $0.2180 \pm .0117$ |
| **Learned to optimise:** | | | | |
| GLAD[31]* (NMSE) | $1.1230 \pm .1756$ | $1.1365 \pm .1549$ | $1.4231 \pm .2625$ | $0.9999 \pm .0001$ |
| Deep-graph** [4] | $0.8423 \pm .0026$ | $0.8179 \pm .0269$ | $0.8931 \pm .0103$ | $0.8498 \pm .0017$ |
| **Proposed:** | | | | |
| Recurrent Unrolling | $0.3018 \pm .0080$ | $0.2885 \pm .0075$ | $0.2658 \pm .0108$ | $0.2007 \pm .0081$ |
| Unrolling | $0.1079 \pm .0047$ | $0.0898 \pm .0067$ | $0.1199 \pm .0064$ | $0.1028 \pm .0073$ |
| L2G | $\mathbf{0.0594 \pm .0044}$ | $\mathbf{0.0746 \pm .0042}$ | $\mathbf{0.0735 \pm .0051}$ | $\mathbf{0.0513 \pm .0060}$ |

* For GLAD, we report test NMSE instead of GMSE so as to follow their original setting in [31].
  GLAD is sensitive to the choice of $\sigma^2$ when generating samples (see Figure 2).
** GMSE may not favour Deep-graph as it learns binary structures only (see AUC in Table 2).

community-like, and small-world structural properties respectively. The parameters of each network model are chosen to yield an edge density in $[0.05, 0.1]$. Edge weights are drawn from a log-normal distribution $\log(\mathbf{W}_{ij}) \sim \mathcal{N}(0, 0.1)$. The weighted adjacency matrix is set as $\mathbf{W} = (\mathbf{W} + \mathbf{W})^\top/2$ to enforce symmetry. We then generate graph-structured Gaussian samples (similar to [23]) with

$$\mathbf{X} \sim \mathcal{N}(\mathbf{0}, \mathbf{K}^{-1}), \ \mathbf{K} = \mathbf{L} + \sigma^2 \mathbf{I}, \ \text{where } \sigma = 0.01. \tag{9}$$

The out-of-sample normalised mean squared error for graph recovery (GMSE) and area under the curve (AUC) are reported to evaluate the prediction of edge weights and the recovery of the binary graph structure. Specifically, GMSE $= \frac{1}{n} \sum_{i=1}^{n} ||\hat{\mathbf{w}}_i - \mathbf{w}_i||_2^2 / ||\mathbf{w}_i||_2^2$. The mean and 95% confidence interval are calculated from 64 test samples.

We include the following baselines in the comparison: 1) **primal-dual splitting algorithm** [16, 21] that is unrolled into a neural network in our model; 2) the iterative algorithm of **ADMM** (details in Appendix). For PDS and ADMM, we tune the hyperparameters on training set via grid search based on GMSE. For the state-of-the-art models that share the idea of learning to optimise, we compare against 3) **GLAD**[2] [31] and 4) **Deep-graph**[3] [4] that are previously introduced in Section 4. Besides the proposed 5) **learning to learn graph topologies (L2G)** framework that minimises the objective in Eq.(8) to obtain a data-to-graph mapping, we also evaluate the performance of ablation models without TopoDiffVAE in replacement of the proximal operator. This model is equivalent to unrolling PDS directly while allowing parameters varying across the layers, which is referred to as 6) **Unrolling** in the following sections. We also consider 7) **Recurrent Unrolling** with shared parameters across the layers, i.e $\alpha^{(0)} = \alpha^{(1)} = \cdots \alpha^{(T-1)}$, $\beta^{(0)} = \beta^{(1)} = \cdots \beta^{(T-1)}$ and $\gamma^{(0)} = \gamma^{(1)} = \cdots \gamma^{(T-1)}$. The loss function for Unrolling and Recurrent Unrolling reduces to the first term in Eq.(8), i.e. without the KL divergence term of TopoDiffVAE. More implementation details, e.g. **model configurations** and **training settings**, can be found in Appendix. We also release the code for implementation [4].

**Ability to recover graph topologies with specific structures.** We evaluate the accuracy of recovering groundtruth graphs of different structural properties from synthetic data. We train an L2G for each set of graph samples that are generated from one particular random graph model, and report GMSE between the groundtruth graphs and the graph estimates on test data in Table 1.

The proposed L2G reduces the GMSE by over 70% from the iterative baseline PDS on all types of graphs. Significant drops from PDS to Recurrent Unrolling and to Unrolling are witnessed. The scenario of recovering the scale-free networks sees biggest improvement on GMSE, as they are relatively hard to learn with handcrafted iterative rules. As shown in Figure 9 in Appendix, the power-law degree distribution associated with such networks is preserved in graph estimates from L2G, while PDS and Unrolling have difficulties to recognise small-degree nodes and nodes that are connected to many other nodes, respectively. PDS and Deep-Graph also fail the KS test, which

---

[2]https://github.com/Harshs27/GLAD

[3]https://github.com/eugenium/LearnGraphDiscovery

[4]https://github.com/xpuoxford/L2G-neurips2021

Table 2: Structural Metrics of recovered binary graph structure

| metric / model | groundtruth | PDS | Deep-Graph [4] | Unrolling | L2G |
|---|---|---|---|---|---|
| **Scale-free (BA):** | | | | | |
| AUC | - | 0.934±.009 | 0.513±.017 | 0.993±.001 | **0.999±.000** |
| KS test score* | 95.31% | 65.63% | 59.38% | 93.75% | **95.31%** |
| **Community (SBM):** | | | | | |
| AUC | - | 0.926±.006 | 0.586±.021 | 0.989±.002 | **0.993±.002** |
| community score | 0.463±.002 | 0.481±.002 | 0.343±.006 | 0.458±.001 | **0.469±.003** |
| **Small-world (WS):** | | | | | |
| AUC | - | 0.913±.007 | 0.529±.010 | 0.984±.005 | **0.991±.000** |
| average shortest path | 2.334±.028 | 2.656±.204 | 1.136±.008 | 2.328±.040 | **2.331±.041** |
| clustering coefficient | 0.323±.018 | 0.514±.035 | 0.906±.004 | 0.433±.016 | **0.382±.014** |

\* KS test score is the percentage of test graphs whose degree sequence passes the Kolmogorov -Smirnov test for goodness of fit of a power-law distribution, i.e. $p$-value > 0.05.

measures how close the degrees of graph estimates follow a power-law distribution (see Table 2 where a $p$-value less than 0.05 rejects the null hypothesis that the degrees follow a power-law distribution). We also test the models on SBM and WS graphs. Please see Appendix C.2 for analysis.

**Comparisons with SOTA.** GLAD[31] and Deep-graph[4] are the few state-of-the-art learning-based methods worth comparing to, but there are notable differences in their settings that affect experimental results. As discussed in Section 4, the objective of GLAD[31] is to learn the precision matrix where the diagonal entries are considered. The output of Deep-graph[4] is an undirected and unweighted (i.e. binary) graph structure. By comparison, the product of generic graph learning adopted in our work is a weighted adjacency matrix, where the diagonal entries are zeros due to no-self-loop convention.

For a fair comparison, we provide the following remarks. Firstly, we stick to their original loss function NMSE for training GLAD[31] and reporting the performance on test data in Table 1. The difference between GMSE and NMSE is that the latter considers the $\ell_2$ loss on diagonal entries of the precision matrix $\Theta$ in GLAD[31]. Note that GMSE equals NMSE when applied to adjacency matrices which are the focus of the present paper. The experimental results show a far inferior performance of GLAD[31] if their NMSE is replaced with our GMSE.

$$\text{GMSE} = \frac{1}{n}\sum_{i=1}^{n}\frac{||\hat{\mathbf{w}}_i - \mathbf{w}_i||_2^2}{||\mathbf{w}_i||_2^2}, \quad \text{NMSE (GLAD[31])} = \frac{1}{n}\sum_{i=1}^{n}\frac{||\hat{\Theta}_i - \Theta_i||_2^F}{||\Theta_i||_2^F} \quad (10)$$

Secondly, a potential limitation of GLAD[31] is that it typically requires good conditioning of the precision matrix, i.e. a large $\sigma^2$ in Eq.(9). On the other hand, in the graph signal processing and graph learning literature [11, 16], smooth graph signals follow a Gaussian distribution where the precision matrix is taken as the graph Laplacian matrix, which is a singular matrix with at least one zero eigenvalue. A large $\sigma^2$ is more likely to destroy the graph topological structure behind the data. Given that our objective is to learn a graph topology with non-negative edge weights, we report results in Table 2 using a small $\sigma = 0.01$. In Figure 2, we see that GLAD[31] performs badly with small diagonal perturbation $\sigma^2$, while our L2G method performs stably with different $\sigma^2$. Admittedly, GLAD[31] outper-

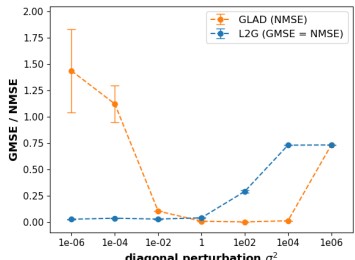

Figure 2: GMSE v.s. $\sigma^2$ for L2G/GLAD.

forms L2G in terms of GMSE/NMSE under $1 \le \sigma^2 \le 10^4$ in Figure 2. However, this could be due to the fact that during training GLAD makes use of the groundtruth precision matrix which contains information of $\sigma^2$, while L2G does not require and hence is not made aware of this information.

Thirdly, we generate another synthetic dataset with binary groundtruth graphs to have a fairer comparison with Deep-graph[4] and report the experimental results in Table 3 of Appendix C.1. We follow the original setting in the paper for training and inference. For both binary graphs or weighted

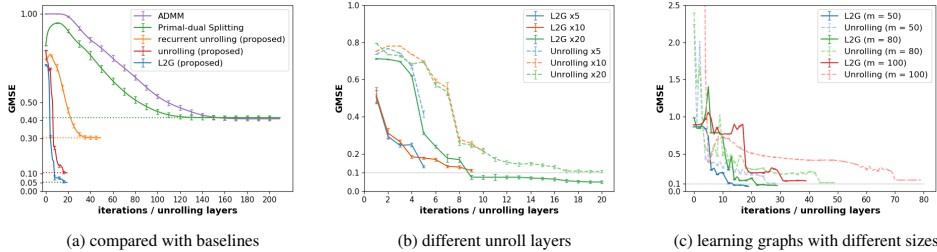

|     |     |     |
| :-: | :-: | :-: |
| (a) compared with baselines | (b) different unroll layers | (c) learning graphs with different sizes |

Figure 3: Convergence of number of iterations

graphs, Deep-graph[4] has achieved inferior performance, showing that CNN has less expressiveness than the inductive bias introduced by model-based graph learning.

**Analysis of model behaviour.** Figure 3a shows the proposed models require far less iterations to achieve a big improvement on accuracy than the baseline models. Here, the number of iterations is referred to as the number of unrolling layers $T$ in Algorithm 2. This means the proposed models can be used to learn a graph from data with significant saving in computation time once trained. In Figure 3b, we vary $T$ in both L2G and Unrolling. With the same $T$, L2G with TopoDiffVAE replacing the proximal operator in Unrolling achieves a lower GMSE at each layer. This suggests that TopoDiffVAE can help the unrolling algorithm to converge faster to a good solution (in this case a graph with desired structural properties).

In Figure 3c, we train L2G and Unrolling on different sizes of graphs, where $m$ is the number of nodes. A larger graph requires more iterations to converge in both models, but the requirement is less for L2G than for Unrolling. One advantage of applying Unrolling is that it can be more easily transferred to learn a graph topology of different size. This is because the parameters learned in Unrolling are not size-dependent. Figure 3c shows that with a large number of iterations (which will be costly to run), Unrolling can learn a effective data-to-graph mapping with a little sacrifice on GMSE. However, L2G significantly outperforms Unrolling in terms of preserving topological properties, which are hard to be reflected by GMSE. Unrolling might fail at detecting edges that are important to structural characteristics. For example, as shown in Figure 8c and 8d in Appendix, rewired edges in WS graphs are missing from the results of Unrolling but successfully learned by L2G. In Appendix C, we discuss these results as well as scalability in more details.

## 6 Real-world applications

Many real-world graphs are found to have particular structural properties. Random graph models can generate graph samples with these structural properties. In this section, we show that L2G can be pretrained on these random graph samples and then transferred to real-world data to learn a graph that automatically incorporates the structural priors present in training samples.

**S&P 100 Stock Returns.** We apply a pretrained L2G on SBM graphs to a financial dataset where we aim to recover the relationship between S&P 100 companies. We use the daily returns of stocks obtained from YahooFinance[5]. Figure 4 gives a visualisation of the estimated graph adjacency matrix where the rows and columns are sorted by sectors. The heatmap clearly shows that two stocks in the same sectors are likely to behave similarly. Some connections between the companies across sectors are also detected, such as the link between CVS Health (38 CVS) and Walgreens Boosts Alliance (78 WBA). This is intuitive as both are related to health care. More details can be found in Appendix.

**Assistant Diagnosis of Autism.** L2G is pretrained with sparse ER graphs and then transferred to learn the brain functional connectivity of Autism from blood-oxygenation-level-dependent (BOLD) time series. We collect data from the dataset[6], which contains 539 subjects with autism spectrum

---

[5]https://pypi.org/project/yahoofinancials/

[6]http://preprocessed-connectomes-project.org/abide/

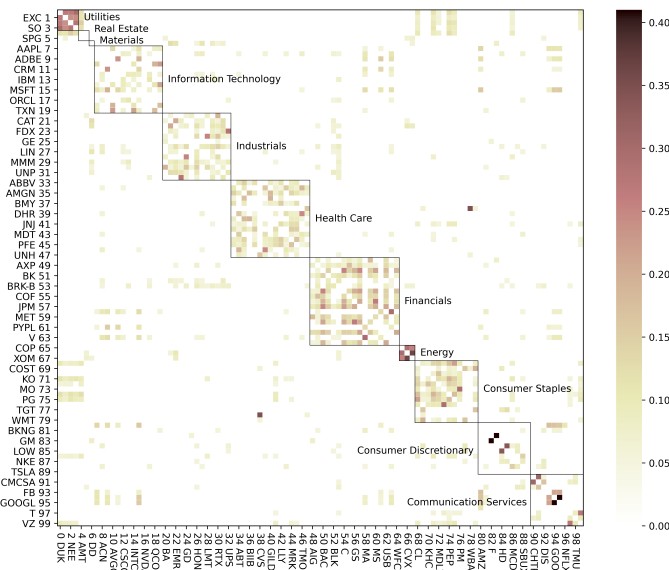

Figure 4: The learned graph adjacency matrix by L2G that reveals the relationship between S&P 100 companies. The rows and columns are sorted by sectors.

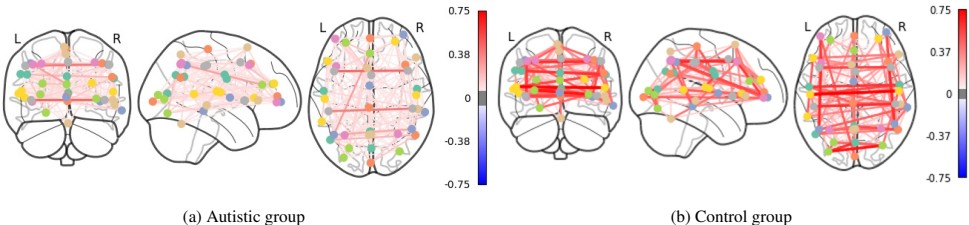

(a) Autistic group          (b) Control group

Figure 5: The connectivity of the 39 regions in brain estimated by L2G using 35 subjects.

disorder and 573 typical controls. We extract BOLD time series from the fMRI images by using atlas with 39 regions of interest [35]. Figure 5 shows that the learned connectivity of the 39 regions using data from 35 subjects in the autistic group (Figure 5a) and 35 subjects in the control group (Figure 5b). The functional connectivity of two groups are apparently different, which provides reference for judging and analysing autism spectrum disorder. The difference is also consistent with current studies showing the underconnectivity in ASD [17]. The stability analysis of the recovered graph structure is presented in Appendix, where our method is shown to be more stable than other methods.

## 7 Conclusion

In this paper, we utilise the technique of learning to optimise to learn graph topologies from node observations. Our method is able to learn a data-to-graph mapping, which leverages the inductive bias provided by a classical iterative algorithm that solves structural-regularised Laplacian quadratic minimisation. We further propose a topological difference variational autoencoder to enhance the expressiveness and flexibility of the structural regularisers. Limitations of the current method include its focus on unrolling an algorithm for learning undirected graph, and scalability to learning large-scale graphs. We leave these directions as future work.

## Acknowledgements

X.P. and X.D. gratefully acknowledge support from the Oxford-Man Institute of Quantitative Finance. S.C gratefully acknowledges support from the National Natural Science Foundation of China under Grant 6217010074 and the Science and Technology Commission of Shanghai Municipal under Grant 21511100900. The authors thank the anonymous reviewers for valuable feedback, and Ammar Mahran, Henry Kenlay and Yin-Cong Zhi for their helpful comments.

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
