# OpenReview forum: "Learning to Learn Graph Topologies"
_NeurIPS.cc/2021/Conference — NeurIPS 2021 Poster_

### Official Review · Reviewer_H66k · 2021-07-16

**Rating:** 7
**Confidence:** 4

**Summary:**

This paper presents a novel method to estimate a graph topology between nodes directly from observations on nodes. In the field of graph signal processing and graphical models, it has been shown that we can estimate the graph topology from data observations at nodes when we can assume that data values at nodes vary smoothly over a graph, in other words when data values observed at node A would be statistically close to the data values observed at node B if node A and B are located in close positions at the entire graph.

This assumption enables us to formulate the graph topology estimation as an optimization problem over the space of graph candidates. In this formulation, the objective function often is defined as a convex function having a form of "graph-data fidelity term" + "structural regularization term", which is then optimized by many iterative algorithms such as proximal gradient descent.

This paper aims to write these handcrafted convex regularizers by a differentiable module, and perform end-to-end learning in a more expressive and flexible manner. This can relax the strong assumption required for each specific regularization (such as L1 or L2 penalty), and extend the same framework flexibly to a wider range of graphs, scale-free graphs, or small-world graphs, which is usually difficult to explicitly write out the corresponding relevant regularization.

A very interesting technical point is, the paper's approach is different from just replacing the regularizer with a neural network module, rather enroll the optimization steps of proximal gradient descent as an iterative procedure, and only critical parts are redesigned by neural networks. By doing this, the method can leverage the effective inductive bias developed in the field not to over-reformulate that target problem.

The experimental evaluations on both synthetic and real datasets demonstrate that this idea actually worked nicely, also showing clear superiority over existing approaches such as GLAD and DeepGraph.

**Limitations And Societal Impact:**

It is appreciated to describe the limitations on the applicable scope of this proposed method.

**Main Review:**

This paper presents a very interesting and principled way to the network topology estimation from data on nodes as an optimization form of eq (1). Usually, we need to design the regularization term, Omega(L), in eq(1) by domain knowledge, but because we need to mathematically solve this optimization as convex optimization, and the options for any possible forms for Omega can be limited. Thus, the idea of enrolling the iterative procedure of optimization, and redesign some critical parts by a flexible neural network sounds quite interesting and promising. Also, it was nice to empirically see that this idea worked in their experimental evaluations.

Overall I liked the idea, and but here are some questions and comments.

- It is quite interesting that in the original enrolled description of proximal-dual splitting at Algorithm 1, only the proximal projection for the primal variable (line 5, Algorithm 1) was replaced by a VAE. The paper explained this motivation as "We do not consider a trainable function to replace the proximal operator in Step 6 of Algorithm 1, as it projects dual variables that do not have a direct connection to graph structures."
However, if the dual variables correspond to the node degree, it should be also highly relevant to the graph structure. Did you actually try also replacing the dual variable with some NN and saw a poor performance or this is just an assumption?
Originally both Omega_1 for primal and Omega_2 for dual come from the "handcrafted structural priors" as in line 112. So relaxing both of them sounds more natural at the first glance, given the purpose of the paper explained. The current form is a kind of hybrid mixture of "handcrafted" + "neural", and I'd like to understand this point more in detail. What does it really mean this leaving 2nd regularizer as it is? If we go neural for both of them, the behavior as a proximal gradient descent became chaotic...?

- In Algorithm 1, why line 5 is gamma-free but line 6 is not? I suppose that line 5 should also come with gamma (somewhere around in "max{0,r_1(t)}"), but either way, we replace it a VAE, and doesn't matter...?

- Related to the question above, I'd like to ask the scope of the proposed approach. We see that it worked for synthetic cases of many variations of random graphs such as BA, ER, SBM, and WS including many model forms of real-world graphs. But, the current form should be affected by "the restriction" of leaving the 2nd regularizer as is. How can this design policy affect the applicable scope of the proposed method? Either way, we need to have some assumptions for the relationship between graph topology and observations on nodes, and is there some guideline on what kinds of graphs can be estimated from observations on nodes by this proposed method?


**Time Spent Reviewing:**

3 hours

---

> ### Author Response · Authors · 2021-08-10
> **Response to Reviewer H66k's comments**
>
> We thank the Reviewer for spending time reading the paper and their positive feedback. We found the Reviewer's comments very interesting and important for us to further improve our work. The followings are our replies to the detailed questions by bullet points.
>
> -----------
> **Q1\.** Can we replace the proximal projection of $\Omega_2$ with some neural networks as well?
>
> **R1\.** Your comments on relaxing $\Omega_2$, i.e. replacing Step 6 in Algorithm 2 is very interesting and we did examine this before.
>
> (1a) Firstly, we would like to elaborate on the connection between dual variables and node degree.  The log regulariser in Eq.(4) indeed operates on node degrees. However, it should be noted that in Algorithm 1, $v$ is the dual variable which is not equal to the vector of node degrees. We agree with the reviewer that the dual variable $v$ is associated with node degree. However, we find it difficult to use a neural network to approximate such an association. Moreover, it is not easy to find a loss surrogate between $v$ and the node degree vector due to the fact that they are not equal.
>
> (1b) Despite the argument above, we did try to replace Step 6 with a simple neural network, e.g. several layers of MLP. It did not become chaotic, but the performance cannot beat Unrolling. We speculate such an architecture with hundreds of neurons might be less expressive in this case and perplex the learning process. It would be very interesting to find a neural network architecture that is expressive enough to capture the association between the node degree vector and dual variable $v$. This is one of the future directions we consider.
>
> -----------
>
> **Q2\.** In Algorithm 1, why Line 5 is gamma-free but Line 6 is not?
>
> **R2\.** It is because during the mathematical derivation, $\gamma$ is cancelled out for Line 5 while not cancelled out for Line 6. The derivation is as follows:
>
> Line 5 should be $p_1^{(t)} = prox_{\gamma \Omega_1}(r_1^{(t)})$ from Algorithm 6 in [19]. $\Omega_1(r_1^{(t)}) = 0$ if $r_1^{(t)} \geq 0$ else $\Omega_1(r_1^{(t)}) = \infty$. So, $\gamma \Omega_1(r_1^{(t)}) =  \Omega_1(r_1^{(t)})$. Hence, $prox_{\gamma \Omega_1}(r_1^{(t)}) = prox_{\Omega_1}(r_1^{(t)}) = \max( 0, r_1^{(t)} )$.
>
> Line 6 should be $p_2^{(t)} = prox_{\gamma \Omega^{\ast}_2}(r_2^{(t)})$ from Algorithm 6 in [19], where $\Omega^{\ast}_2$ is the conjugate of $\Omega_2$. With the Moreau decomposition, we can derive the proximal operator of the conjugate such that
>
> $$prox_{\gamma \Omega^{\ast}_2} (r_2^{(t)}) $$
>
> $$ = r_2^{(t)} - \gamma prox_{\frac{1}{\gamma} \Omega_2}(\frac{r_2^{(t)}}{\gamma})$$
> $ =  \frac{r_2^{(t)} - \sqrt{r_2^{(t)2} + 4 \alpha \gamma}}{2} $.
>
> The proof follows:
> Denote $h(r_2^{(t)}) = \gamma \Omega_2(\frac{r_2^{(t)}}{\gamma})$. We have $$h(r_2^{(t)})
> = \gamma (  -\alpha 1^T \log(\frac{r_2^{(t)}}{\gamma}) )
> =\gamma ( -\alpha 1^T \log(r_2^{(t)}) + \alpha 1^T\log \gamma)
> = \gamma ( -\alpha 1^T \log(r_2^{(t)}) + \text{constant})
> = \gamma (\Omega_2(r_2^{(t)})+ \text{constant}).$$
>
> Consequently, $prox_{h}(r_2^{(t)}) = prox_{\gamma \Omega_2}(r_2^{(t)})$, with the affine rule of proximal operators.
>
> Meanwhile, $prox_{\gamma \Omega_2}(r_2^{(t)}) = \frac{r_2^{(t)} + \sqrt{r_2^{(t)2} + 4 \alpha \gamma}}{2}$ (see proof on Page 136 of [First-order Methods in Optimization by Amir Beck](https://archive.siam.org/books/mo25/mo25_ch6.pdf) ).
> By Theorem 6.12 on Page 138 of the book, we also have that if $h(r_2^{(t)}) = \gamma \Omega_2(\frac{r_2^{(t)}}{\gamma})$, then $prox_{h}(r_2^{(t)}) = \gamma prox_{\frac{1}{\gamma} \Omega_2}(\frac{r_2^{(t)}}{\gamma})$. We then have that $prox_{h}(r_2^{(t)}) = prox_{\gamma \Omega_2}(r_2^{(t)}) = \gamma prox_{\frac{1}{\gamma} \Omega_2}(\frac{r_2^{(t)}}{\gamma})$.
> Plugging it back to the Moreau decomposition leads to
> $$prox_{\gamma \Omega^{\ast}_2}(r_2^{(t)}) $$
>
> $$ = r_2^{(t)} - \gamma prox_{\frac{1}{\gamma} \Omega_2}(\frac{r_2^{(t)}}{\gamma}) $$
> $$ = r_2^{(t)} - prox_{\gamma \Omega_2}(r_2^{(t)}) $$
> $$ = r_2^{(t)} -  \frac{r_2^{(t)} + \sqrt{r_2^{(t)2} + 4 \alpha \gamma}}{2} $$
> $$ =  \frac{r_2^{(t)} - \sqrt{r_2^{(t)2} + 4 \alpha \gamma}}{2}, $$
> which is Line 6 in Algorithm 1. (Q.E.D)
>
> -----------
>
> **Q3\.** Is the scope of the proposed approach restricted to certain graph types?
>
> **R3\.** We do not think there is a restriction as to which topological properties the proposed method can or cannot learn. Given that we learn the topological features from graphs that have been fed into our model, it should apply to a variety of graphs.
> The log regulariser in Eq.(4) is mainly added to prevent all-zero solutions for edge weights, which as suggested in [16] also promotes the connectivity of the graph. We believe its impact is small on the generalisation capability of the proposed method. Even if it does promote connectivity and hence may not favour graphs with isolated nodes, recall that the penalty parameter $\alpha$ is learnable. Our algorithm can therefore make it close to zero if the log regulariser is not effective in certain scenarios. Nevertheless, as mentioned in our response to the first comment of the reviewer, we will explore the possibility of an alternative to this second regulariser as part of future work.
>
> -----------
>
> We hope the above responses explain away your confusion and concerns. Thank you for spending time reading our paper and the responses.

---

> > ### Comment · Reviewer_H66k · 2021-08-30
> > **Thanks for the clafirication**
> >
> > I have read the authors' response, and would like to thank the authors for the detailed information. It answered my questions raised in the initial review.

---

### Official Review · Reviewer_nzro · 2021-07-16

**Rating:** 6
**Confidence:** 3

**Summary:**

The authors study graph structure learning. The paper's main contribution is to improve the unrolling primal-dual iterative algorithm used to solve equation 4 in the paper. The authors propose a Topological difference variational autoencoder (TopoDiffVAE), which can learn a mapping between two graph domains where one graph domain consists of richer structure information.

**Limitations And Societal Impact:**

Yes

**Main Review:**

Originality
I think there is a lot of novelty in the paper. Though the paper is motivated by some recent works on L2O, according to the authors, it is a very early attempt to address the difficult challenge of designing convex functions to represent complex topological priors.

Quality
1. The authors show superior performance on a few synthetic data sets. According to table 1 and table 2, the L2G's prediction is very much to the ground truth graph structure than other methods. However, I did not see the authors comparing with baselines on real-world applications. For figure 4, can Graphical lasso/group graphical lasso obtain something similar?

2. According to figure 3, can we conclude that the difference between unrolling and L2G (with TopoDiffVAE) can be tiny if we use more iterations? I think simply saying L2G converges faster does not make sense, as the computation price is much higher than the unrolling per iteration.

3. I also have a slight concern about the scalability of the methods. Can the technique be applied to modeling a much larger graph (e.g., >=1000 nodes)?

4. What are the tao values (introduced in equation 8) and your beta value (KL divergence term weight) in your experiments?




Clarity

a. Line 157, typo, w_x, and w_x.

b. Actually, I am a bit confused about what is w_x and w_y.
According to line 5 of Algorithm 2, TopoDiffVAE takes r_1^{t} as input. According to lines 177-178, it seems w_x is r_1^{t} after the gradient descent step, which is line 3 in algorithm 2.

c. For w_y, is there a real ground truth w_y?
Or, according to lines 177-178, it seems w_y takes the form of p_1^{t}? Should it be p_1^{t-1}? Or should it be p_1^{t} in line 5 of algorithm 1?

d. Is there only one pair of w_x and w_y used to train the autoencoder in every of the T steps? I am not sure how the proposed approach would work, given a much larger graph.


Significance
Modeling graph structure is with sufficient interest in both the machine learning community and real-life application.
Though I have a few concerns (it could be I missed something), but the idea of this paper seems to be interesting.

Post rebuttal:  I read the authors's response. The authors have answered all my questions, clarified some small typos and addressed my concerns. Score increased.

**Time Spent Reviewing:**

7

---

> ### Author Response · Authors · 2021-08-10
> **Response to Reviewer nzro 's comments**
>
> We thank the Reviewer for their appreciation of the novelty and overall quality of our submission. We are grateful for their comments on various aspects of the work, which we have tried to address to the best of our capability. We hope that in light of this the Reviewer would consider increasing their score.
>
> ### Quality
> ---------
>
> **Q1\.** Did we compare to baselines on real-world applications?
>
> **R1\.** In our first submission, we show two real-world examples, i.e. one on the finance data and the other on brain functional connectivities.
>
> For the experiment on brain functional connectivities, we did compare it to two baselines: Graphical Lasso and Deep-Graph [3]. Due to the page limitation of the main submission, we put the results and analysis in Appendix (Line 513 - 529 with Table 3) in the supplementary material. Following the same experimental setup in Deep-Graph [3], the comparison is mainly to show the stability and robustness of learned brain functional graphs, which is important for clinical interpretation and applications. The stability and robustness results validate that the proposed method outperforms Graphical lasso and Deep-Graph [3].
>
> For the experiment on finance data, there is no groundtruth, making it not easy to find a quantitative measure to compare model candidates. However, based on domain knowledge in finance, we could choose to visualise the results and give a legitimate interpretation of the learned graph with respect to sector classification. For example, in Figure 4, both the proposed method and graphical Lasso can learn similar graph matrices that show the relationships between the S&P 100 companies. However, the precision matrix learned by graphical Lasso captures both positive and negative partial correlations between the data entities, while the learned graph from our model does not contain any negative edge weight. This non-negative property makes the latter one easier to apply the learned graph in downstream tasks, such as community detection.
>
> ------------------
>
> **Q2\.** According to Figure 3, can we conclude that the difference between unrolling and L2G (with TopoDiffVAE) can be tiny if we use more iterations?
>
> **R2\.** You are certainly right that if we use more iterations, the difference of GMSE between Unrolling and L2G (Unrolling+TopoDiffVAE) is not significant. However, L2G significantly outperforms Unrolling in terms of preserving topological properties, which are hard to be reflected by GMSE. Unrolling might fail at detecting edges that are important to structural characteristics. For example, as shown in Figure 8(c) and 8(d), rewired edges in WS graphs are missing from the results of Unrolling but successfully learned by L2G. In addition, as shown in Figure 9(c) and 9(d), the degree distribution is less well preserved from Unrolling. The GMSEs in both cases, however, are similar and both relatively small.
>
> Please refer to the Appendix in the supplementary material for Figure 8 and Figure 9 mentioned above.
>
> ------------------
>
> **Q3\.** Can the technique be applied to modeling a much larger graph (e.g., $\geq$ 1000 nodes)?
>
> **R3\.** Regarding the scalability concern, applying L2G or Unrolling to a graph with thousands of nodes is possible. Generally, the computational complexity is quadratic in the graph size and also varies according to the configurations of TopoDiffVAE in L2G.
>
>
> The theoretical complexity for L2G is $\mathcal{O}(TB(m^2n_1 + m^2n_4 + n_1^2m + n_1n_2 + n_2n_3 + n_3n_4))$, where $T$ is the number of unrolling layers, $B$ is the batch size of training samples, $n_1$, $n_2$, $n_3$ and $n_4$ are labelled as nhid, emb\_out, nlatent and nhid2 in Figure 6 (please refer to Line 471-474 in Appendix in the supplementary material for their definitions). The theoretical complexity for Unrolling and Recurrent Unrolling is $\mathcal{O}(TBm^2)$. Note that Unrolling requires more iterations (i.e., $T$) to converge than L2G. By comparison, the theoretical complexity for the iterative baselines, e.g. PDS[16], is $\mathcal{O}(m^2)$ for each iteration (see Table 1 of Mateos et al. 2020).
>
>
> Meanwhile, the theoretical memory required for storing parameters in L2G is $\mathcal{O}(T(m^2n_4 + 4n_1^2+2n_1n_2 + 2n_2n_3 + n_3n_4))$, while for that in Unrolling is $\mathcal{O}(4T)$, which is trivial. Note that learning an L2G model for larger graphs requires more training samples. Figure 7 in Appendix B.2 provides a rough picture. Dynamic loading training samples can save memory.
>
> Mateos et al. 2020. "Connecting the Dots: Identifying Network Structure via Graph Signal Processing". IEEE Signal Processing Magazine.
>
> --------------------
>
> **Q4\.** What are the tao values (introduced in equation 8) and your beta value (KL divergence term weight) in your experiments?
>
> **R4\.**  We chose tau = 0.9 and beta = 1 for our experiments. We have added this detail in the revised version.
>
> ---------
>
>
>
> ### Clarity
> Regarding the questions on clarify, we realised several points were not described clearly and please find our response below.
>
> ---------
>
> **Q5\.** Line 157, typo, w_x, and w_x.
>
> **R5\.** We have corrected it.
>
> --------------------
>
> **Q6\.** What are w_x and w_y.  According to line 5 of Algorithm 2, TopoDiffVAE takes r_1^{t} as input. According to lines 177-178, it seems w_x is r_1^{t} after the gradient descent step, which is line 3 in algorithm 2.
>
> **R6\.**  Firstly, we used $w_x$ and $w_y$ in Section 3.2 which introduces TopoDiffVAE because we want to have a rigorous introduction that follows the VAE literature. That is to say, $w_x$ is a random variable and $r_1^{(t)}$ is its realisation. So, you are correct that $w_x$ is $r_1^{(t)}$ after the gradient descent step, which is Line 3 in Algorithm 2 (Line 177-178).
>
> --------------------
>
> **Q7\.** For w_y, is there a real ground truth w_y? Or, according to lines 177-178, it seems w_y takes the form of p_1^{t}? Should it be p_1^{t-1}? Or should it be p_1^{t} in line 5 of algorithm 1?
>
> **R7\.** We apologies for being not very clear in Line 177-178 about $w_y$ taking the form of $p_1^{(t)}$. $w_y$ is a random variable whose realisation is the groundtruth edge weights. We mistakenly used the same notation $w_y$ in Eq.(7); the correct output of Eq.(7) should be the estimation of $w_y$. We have changed the notation to $\hat{w}_y$ in the revised version. In Algorithm 2, $r_1^{(t)}$, $p_1^{(t)}$, $q_1^{(t)}$, $w^{(t)}$ and $w^{(t+1)}$ are primal variables that all correspond to edge weights. Therefore, when we mentioned the output of TopoDiffVAE, i.e. $\hat{w}_y$, takes the form of $p_1^{(t)}$, we meant it projects the less structured graph estimation $r_1^{(t)}$ to a better graph estimation $p_1^{(t)}$ that has more similar topological structure to the groundtruth $w_y$. The groundtruth $w_y$ is only used in training procedure, i.e. Eq.(5a) for computing latent variable and Eq.(8) for supervision. During validation or inference procedure, the latent variable is sampled from $\mathcal{N}(0,1)$ in replacement of Eq.(5a) and Eq.(5b). We have changed the corresponding sentence to make it clear in the revised version.
>
> --------------------
>
> **Q8\.** Is there only one pair of w_x and w_y used to train the autoencoder in every of the T steps? I am not sure how the proposed approach would work, given a much larger graph.
>
> **R8\.** Our code is batch-friendly, so we use a batch size of 32 pairs of graphs and data for training. Your intuition is absolutely right, the training size varies according to the graph size (see Figure 7 in Appendix and Appendix B for more details). If you have further concern, please check and try our code attached in the supplementary material.
>
> --------------------
>
> Finally, we would like to thank you again for your valuable comments that our paper is largely benefited from.

---

### Official Review · Reviewer_aKqR · 2021-07-17

**Rating:** 6
**Confidence:** 4

**Summary:**

This work studies how to learn the graph topology, the algorithm is developed based on learning to optimize (L2O), which first unrolls an iterative primal-dual splitting algorithm into a neural network, and then the network is stacked with a variantional autoencoder that refines the estimated graph with structure properties. Experiments on both synthetic and real-world data demonstrate the model is more efficient different iterative algorithms.

**Limitations And Societal Impact:**

Yes.

**Main Review:**

The reviewer finds the paper clearly written and easy to follow, but the originality and significance need to be further clarified.

One major concern of the reviewer is that the model is adopting the L2O, and it is not clear whether the effectiveness is due to the  L2O or due to the modeling and regularization proposed in this work. Note that it is known that L2O provides an advantage compared with ADMM and PDS compared in this work.

While the paper indeed compared with other L2O algorithms including [29] and [3]. But the reviewer finds the performance of these two algorithms surprisingly unsatisfactory in Table 1, with GMSE close to 1, and is significantly worse than conventional learning algorithms including ADMM and PDS. This does not seem right to the reviewer. For example, as is pointed out by the author and can be seen from Figure 2, the performance of GLAD in fact outperforms the proposed L2G if sigma^2 is selected as 1 and improves with a larger sigma value. However, Figure 2 only shows sigma^2 from 1e-6 to 1 which is a very small range. it is likely that the performance will be stable if sigma improves even more and the performance will be stable if sigma>1, which is not shown in figure 2.  Similarly Comparing deep-graph which is designed for binary graph for a weighted synthetic graph is not a fair comparison. Hence, the experiments cannot convince the reviewer the proposed algorithm indeed outperform existing L2O-based algorithms since the parameter selection and the generation of the synthetic data might favor the current scheme.

Similar concerns exist for the experiments in Table 2, as the alternative L2O-based algorithm achieves worth result than the basic primal-dual splitting algorithm (PDS) which is surprising and may be due to an unfavorable selection of parameters.

The experiments on real-datasets seem promising, but given the concerns mentioned above, the reviewer is not very convinced by the comparison.


-------

Post rebuttal: The reviewer appreciates the authors for their efforts in addressing the questions and providing more thorough and convincing numerical results. The major concerns have been addressed and the score has been raised.

**Time Spent Reviewing:**

2.5

---

> ### Author Response · Authors · 2021-08-10
> **Response to Reviewer aKqR's comments**
>
> We thank the Reviewer for spending time reading our work. We are grateful for the insightful comments on the L2O methodology and the comparison with existing models in the literature. Below we have tried to address the Reviewer's concerns to the best of our capability. We hope that in light of this the Reviewer would consider increasing their score.
>
> ------------
> ### The effectiveness of the proposed model
>
> **Q1\.** One major concern of the reviewer is that the model is adopting the L2O, and it is not clear whether the effectiveness is due to the L2O or due to the modeling and regularization proposed in this work.
>
> **R1\.** In your first major concern, we assume that the 'L2O' indicates the proposed algorithmic unrolling for graph learning, the 'modeling' here means the proposed objective in Eq.(4) and the 'regularisation' means the proposed TopoDiffVAE in replacement of the original proximal operator of a regulariser. The short version of our response is that the proposed algorithmic unrolling framework, the proposed objective in Eq.(4), and the proposed TopoDiffVAE all contribute to better performance and effectiveness.
>
> It is also worth clarifying the difference between L2O and L2G. As mentioned in the latest survey of L2O [7], L2O is recognised as a general concept that involves algorithmic unrolling and deep prior learning. On the other hand, the proposed L2G is a specific algorithm that leverages both the algorithmic unrolling framework and TopoDiffVAE to achieve graph learning.
>
> To validate the effectiveness of the proposed TopoDiffVAE, we compare the unrolling framework with and without TopoDiffVAE (i.e named 'L2G' and 'Unrolling' in our paper). The experimental results show that the unrolling without TopoDiffVAE achieves an error (GMSE) of $0.108 \pm .005$; the unrolling with TopoDiffVAE achieves $0.059 \pm .004$. Please refer to Table 1, Table 2 and Figure 3(a) for more details. These comparisons reflect that the proposed TopoDiffVAE brings significant benefits.
>
> To validate the effectiveness of the proposed objective in Eq.(4), we compare Eq.(4) with the objective of Graphical Lasso (i.e. the $l_1$ regularised inverse covariance estimation) given the solver ADMM. The experimental results show that ADMM while solving Eq.(4) achieves a GMSE of $0.409 \pm .005$; ADMM while solving Graphical Lasso achieves $0.422 \pm .005$. Note that GLAD [29] applies algorithmic unrolling to solve Graphical Lasso, whose performance is discussed in the next point (R2) of this response. These comparisons reflect that the proposed objective in Eq.(4) slightly outperforms the objective of Graphical Lasso in terms of learning a graph adjacency matrix with non-negative edge weights. The results are not surprising, as Graphical Lasso aims at learning the inverse covariance. Note that the graph adjacency matrix and inverse covariance are different graph representations, which is well articulated in the graph learning literature [10, 16, 21].
>
> Please refer to our paper for the above references.
>
> ------------
>
> ### The performance of SOTA  (GLAD [29] and Deep-graph [3])
> Regarding your concern about the performance of SOTA (GLAD [29] and Deep-graph [3]) reported in our paper being surprisingly worse, we would like to provide the explanations for GLAD and Deep-graph separately in the following two responses. We are open to further discussions and suggestions from the Reviewer on these comparisons.
>
> ------------
>
> **Q2\.** Why GLAD[29] achieved surprisingly unsatisfactory performance in Table 1? Will it be stable and improve even more if $\sigma^2 > 1$?
>
> **R2\.** In Figure 2, we only plotted the situation when $\sigma^2 \leq 1$ to show that GLAD [29] is sensitive to the values of $\sigma^2$. Here, we add the situation when $\sigma^2 > 1$ in the table below.
>
>
> > A Table for GMSE/NMSE under different $\sigma^2$ for GLAD/L2G.
> >
> > |     $\sigma^2$    |       1e-06      |       1e-04      |       1e-02      |         1        |       1e02       |            1e04            |            1e06            |
> |:----------------------------------|:---------------------|:----------------|:----------------|:----------------|:----------------|:--------------------------|:--------------------------|
> |    GLAD(NMSE)    | $1.437 \pm .393$ | $1.123 \pm .176$ | $0.105 \pm .002$ | $0.009 \pm .001$ | $0.002 \pm .000$ |      $0.013 \pm .000$      |      $0.733 \pm .000$      |
> | L2G(NMSE=GMSE) | $0.058 \pm .005$ | $0.071 \pm .005$ | $0.062 \pm .005$ | $0.060 \pm .001$ | $0.291 \pm .011$ |      $0.731 \pm .000$      |      $0.733 \pm .000$      |
> |  |  | | | | | | |
>
>
> From the results,  we would like to provide the following explanations:
>
> 1. The original paper of GLAD [29] used a slightly different GMSE (they called it NMSE in [29]). Our GMSE is defined over off-diagonal edge weights as $\frac{||\hat{w} - w||^2_2}{||w||^2_2}$, where $w$ includes the lower-triangular elements of the graph adjacency matrix; while their NMSE is defined over the whole precision matrix $\frac{||\hat{\Theta} - \Theta||^F_2}{||\Theta||^F_2}$, where $\Theta$ is the precision matrix. The difference is whether to include the diagonal entries in the evaluation, and this is due to the different objectives of the two methods: while we aim at learning a weighted graph with non-negative edge weights, GLAD aims at learning a precision matrix which contains both positive and negative entries and diagonal entries that have statistical interpretation. Therefore, the two methods are not directly comparable.
>
> 2. More importantly, GLAD has access to the information of $\sigma^2$ during training (this is required by their method) while L2G does not need this information, which leads to an unfair comparison. This is the main reason why, under $1 \leq \sigma^2 \leq 1e04$, GLAD [29] outperforms L2G in terms of GMSE/NMSE in the table above.
>
> 3. As explained in Line 259-266, a potential limitation of the objective function in GLAD, which is designed for the purpose of inverse covariance estimation, is that it typically requires good conditioning of the precision matrix (large $\sigma^2$).
>
>     On the other hand, in the graph signal processing and graph learning literature (see [10, 16]), smooth graph signals follow a Gaussian distribution where the precision matrix is taken as the graph Laplacian matrix, which is a singular matrix with at least one zero eigenvalue. A large $\sigma^2$ in the generative model in Eq.(9) is more likely to destroy the graph topological structure behind the data.
>
>     Given that our objective is to learn a graph topology with non-negative edge weights, our choice of a small $\sigma^2$ to report results in Table 1 ($\sigma=0.01$) better aligns with the graph learning literature.
>
> To allow for a more comprehensive understanding of the comparison, we will add the table above and these discussions to the final version of the paper.
>
> -------------
>
> **Q3\.**  Why Deep-graph [3] achieved surprisingly unsatisfactory performance in Table 1 and Table 2? Is it fair to compare Deep-graph which is designed for binary graphs for weighted synthetic graphs?
>
> **R3\.** We thank the reviewer for raising the point. We would like to provide the following explanations on the performance of Deep-graph[3]:
>
> 1. First, we would like to clarify that Deep-graph [3] is not strictly categorised as L2O in literature [7, 24], since it has a CNN architecture that does not inherit from any classical iterative algorithms. It is therefore more like a baseline for the learning-based method (see Line 202-213).
>
> 2. One reason why Deep-graph achieves worse results than the basic PDS/ADMM in Table 1 and Table 2 is that, similar to GLAD, Deep-graph was designed to solve a different objective to obtain an inverse covariance matrix (a.k.a. precision matrix). As discussed above, our objective in Eq.(4) is more suitable and commonly used in literature in learning graph adjacency matrices with non-negative edge weights (see [10, 16]). Furthermore, as explained in Line 255-258, the performance of Deep-graph shows that a simple CNN architecture has less expressiveness than the inductive bias introduced by model-based graph learning.
>
> 3. Regarding your concern on weighted synthetic graphs that might be a reason why Deep-graph achieves worse performance, we rerun the experiments with synthetic binary groundtruth graphs and the results are as follows.
>
>     >  A Table of GMSE in recovering binary graphs.
>     >
>     > |model/graph type | scale-free | ER | Community | Small-world |
> |------------ | -------------|  -------------| -------------| -------------|
> |Deep-graph[3] | 0.8017 $\pm$ 0.0031| 0.6935 $\pm$ 0.0498 | 0.9013 $\pm$ 0.0158 | 0.8726 $\pm$ 0.0101 |
> |L2G | 0.0603 $\pm$ 0.0023 | 0.0833 $\pm$ 0.0051 | 0.0798 $\pm$ 0.0100 | 0.0559 $\pm$ 0.0035 |
> | | | | | |
>
>     > A Table of Structural Metrics in recovering binary graphs.
>     >
>     > |model/graph type | groundtruth | Deep-graph[3] | L2G |
> |------------ | -------------|  -------------| -------------|
> |Scale-free (AUC) | - | 0.565 $\pm$ .101 | 0.999 $\pm$ .000|
> |Scale-free (KS score) | 96.15\% | 68.32\% | 96.15\% |
> |Community (AUC) | - | 0.701 $\pm$ .088 | 0.992 $\pm$ .001|
> |Community score | 0.472 $\pm$ .002 | 0.311 $\pm$ .006 | 0.479 $\pm$ .005|
> |Small-world (AUC) | - | 0.519 $\pm$ .023 | 0.997 $\pm$ .000|
> |Small-world (average shortest path) | 2.229 $\pm$ .028 | 1.111 $\pm$ .008 |  2.225 $\pm$ .041|
> | | | | |
>
> Note a smaller GMSE means a better performance; and a higher AUC means a a better performance.
>
> In summary, the results in the above two tables can lead to the same conclusion as in Table 1 and Table 2 of our first submission that Deep-graph [3] has a less satisfactory performance. We will add the above tables in the appendix of the revised version.
>
> -------------------
>
> Thank you again for spending time reading our paper and responses.

---

### Official Review · Reviewer_8sof · 2021-07-20

**Rating:** 8
**Confidence:** 4

**Summary:**

This paper aims to learn a graph from observations/signals on data entities. They propose a new method based on Learning to Optimise (L2O) whereby they map the node data to a graph. They argue that more complex phenomena like scale-free and small-world properties cannot simply be modelled or incorporated as priors in convex regularizers that are handcrafted. Instead, they propose an unrolling of a well-known iterative primal-dual splitting algorithm designed specifically for graph learning by a neural network, namely a topological difference variational autoencoder termed TopoDiffVAE. The proposed method improves over the results by a large margin compared with the baselines (iterative solvers for graph learning) in generating graphs from synthetic random graphs and real data.

**Limitations And Societal Impact:**

Yes.

**Main Review:**

-- Strengths:
1. The paper is very well-written and easy to understand. I believe the idea is very interesting and novel and will help incorporate very interesting prior knowledge into graph learning which can represent very complex topological priors. The paper was indeed a pleasure to read and the technical details were clearly elucidated.

2. The L20 method for learning graph structure from signal data proposed by the authors is extremely interesting and quite timely. Especially the idea of taking a well-known iterative optimization algorithm for graph learning and replacing it with an auto-encoder in the right places.

3. Improves over the results by a large margin compared with the baselines in generating graphs from synthetic random graphs and real data.

4. Clearly exhibits the ability to transfer the learnings on random graphs to real world settings


-- Weaknesses:
1. How is the T (number of iterations) in Algorithm 2 ascertained?

2. On Lines 140-142, the authors claim that finding an indicator function where the proximal operator projects onto the space of graphs sharing similar properties is hard to come up with. I would assume that the space of graphs can easily be partitioned after having run them through some kind of a graph kernel (like a WL algorithm maybe?) and then it would just be a matter of defining what a “projection operator” in this graph kernel space might mean? Can this point possibly be expanded upon in the rebuttal?

3. It would be interesting to theoretically analyze the computational complexity of the overhead in terms of the parameters introduced and needed for training in each unrolling step and the resulting scalability issues that arise due to this. I understand this has been mentioned as a future work by the authors, but I think this information would really help get a better understanding of this method.

-- Minor Typos:
Line 148: “abuse use of notation”.

Line 157: w_x repeated.

Line 160: w_x repeated.

Line 210: ...eigenvalues of 1...



**Time Spent Reviewing:**

5 hrs

---

> ### Author Response · Authors · 2021-08-10
> **Response to Reviewer 8sof 's comments**
>
> We thank the Reviewer for the appreciation of and valuable comments on our work. Regarding the comments on the weaknesses, we have scrutinized our work again and address them one by one as follows.
>
> ----------
>
> **R1\.** How is the T (number of iterations) in Algorithm 2 ascertained?
>
> **Q1\.** $T$ is the number of unrolling layers in Algorithm 2, i.e. how many times we repeat Step 3 to Step 10. $T$ is assigned as a hyperparameter in the unrolling networks. We have added it more clearly in the algorithm box in the revised version.
>
> ----------
>
> **R2\.** The possibility of using graph kernels in defining a projection operator.
>
> **Q2\.** Your comment on weakness mentioned a projection operator into a graph kernel space, which is very interesting and might open a new thread for the graph learning problem.
>
> Usually, graph learning aims to construct a graph structure while graph kernel is built based on a fixed graph structure and extracts graph-based features. As far as we are concerned, we could present the high-level topological property of a complicated space of graphs through multiple graph kernels.
>
> In this work, we assume each space of graphs owns a representative topological property, e.g. the scale-free or small-world phenomenon. The proximal operator of $\Omega_1 = \mathcal{I}_{\mathcal{G}}(\cdot)$ projects the graph estimation from data-fidelity term into that space, so the ultimate graph estimation would behave more like the specific topological property, e.g. scale-free networks. In this case, we train different L2G models with different graph types that correspond to different spaces. It could be that partitioning the graph space through some graph kernel provides a way to separate graphs of different topological properties altogether. In this case, further considerations of node permutation, the structure of the kernel space and computational efficiency should be taken into account.
>
> Nevertheless, we believe this is a very interesting direction which we will explore in future work.
>
> ----------
>
> **Q3\.** What is the computational complexity of our method?
>
> **R3\.** Generally, the computational complexity is quadratic in the graph size and also varies according to the configurations of TopoDiffVAE in L2G.
>
> The theoretical complexity for L2G is $\mathcal{O}(TB(m^2n_1 + m^2n_4 + n_1^2m + n_1n_2 + n_2n_3 + n_3n_4))$, where $T$ is the number of unrolling layers, $B$ is the batch size of training samples, $n_1$, $n_2$, $n_3$ and $n_4$ are labelled as nhid, emb\_out, nlatent and nhid2 in Figure 6 (please refer to Line 471-474 in Appendix in the supplementary material for their definitions). The theoretical complexity for Unrolling and Recurrent Unrolling is $\mathcal{O}(TBm^2)$. Note that Unrolling requires more iterations (i.e., $T$) to converge than L2G. By comparison, the theoretical complexity for the iterative baselines, e.g. PDS[16], is $\mathcal{O}(m^2)$ for each iteration (see Table 1 of Mateos et al. 2020).
>
> Meanwhile, the theoretical memory required for storing parameters in L2G is $\mathcal{O}(T(m^2n_4 + 4n_1^2+2n_1n_2 + 2n_2n_3 + n_3n_4))$, while for that in Unrolling is $\mathcal{O}(4T)$, which is trivial. Note that learning an L2G model for larger graphs requires more training samples. Figure 7 in Appendix B.2 provides a rough picture. Dynamic loading training samples can save memory.
>
> Mateos et al. 2020. "Connecting the Dots: Identifying Network Structure via Graph Signal Processing". IEEE Signal Processing Magazine.
>
> ----------
>
> Finally, we have corrected the typos. Thank you so much for spending time reading our work carefully.

---

### Decision · Program_Chairs · 2021-09-27

**Decision:**

Accept (Poster)

**Comment:**

The paper contains much novelty in the formulation of topological constraints and the method of enrolling the iterative procedure of optimization and demonstrates favorable empirical results.  Some reviewers raised unclear points in the paper, but the authors' feedbacks clarified them in detail.  We judge the paper is acceptable in NeurIPS, but we hope the feedback will be reflected in the final version.